# Bone canonical Wnt signaling is downregulated in type 2 diabetes and associates with higher advanced glycation end-products (AGEs) content and reduced bone strength

Giulia Leanza[1,2], Francesca Cannata[1], Malak Faraj[1], Claudio Pedone[3], Viola Viola[1], Flavia Tramontana[1,2], Niccolò Pellegrini[1], Gianluca Vadalà[4], Alessandra Piccoli[1], Rocky Strollo[5], Francesca Zalfa[6,7], Alec T Beeve[8], Erica L Scheller[8], Simon Y Tang[9], Roberto Civitelli[8], Mauro Maccarrone[10,11], Rocco Papalia[4*†], Nicola Napoli[1,2,8*†]

[1]Department of Medicine and Surgery, Research Unit of Endocrinology and Diabetes, Università Campus Bio-Medico di Roma, Via Alvaro del Portillo, Roma, Italy; [2]Operative Research Unit of Osteometabolic and Thyroid Diseases, Fondazione Policlinico Universitario Campus Bio-Medico, Via Alvaro del Portillo, Roma, Italy; [3]Operative Research Unit of Geriatrics, Fondazione Policlinico Universitario Campus Bio Medico, Via Alvaro del Portillo, Roma, Italy; [4]Operative Research Unit of Orthopedic and Trauma Surgery, Fondazione Policlinico Universitario Campus Bio-Medico, Via Alvaro del Portillo, Roma, Italy; [5]Department of Human Sciences and Promotion of the Quality of Life San Raffaele Roma Open University Via di Val Cannuta, Roma, Italy; [6]Predictive Molecular Diagnostic Unit, Pathology Department, Fondazione Policlinico Universitario Campus Bio-Medico, Via Alvaro del Portillo, Roma, Italy; [7]Microscopic and Ultrastructural Anatomy Unit, Università Campus Bio-Medico di Roma, Via Alvaro del Portillo, Roma, Italy; [8]Department of Medicine, Division of Bone and Mineral Diseases, Musculoskeletal Research Center, Washington University School of Medicine, St. Louis, United States; [9]Department of Orthopaedic Surgery, Washington University in St. Louis, St Louis, United States; [10]Department of Biotechnological and Applied Clinical Sciences, University of L'Aquila, Via Vetoio snc, Aquila, Italy; [11]European Center for Brain Research, Santa Lucia Foundation IRCCS, Roma, Italy

*For correspondence:
r.papalia@policlinicocampus.it (RP);
n.napoli@policlinicocampus.it (NN)

†These authors contributed equally to this work

Competing interest: The authors declare that no competing interests exist.

**Abstract** Type 2 diabetes (T2D) is associated with higher fracture risk, despite normal or high bone mineral density. We reported that bone formation genes (*SOST* and *RUNX2*) and advanced glycation end-products (AGEs) were impaired in T2D. We investigated Wnt signaling regulation and its association with AGEs accumulation and bone strength in T2D from bone tissue of 15 T2D and 21 non-diabetic postmenopausal women undergoing hip arthroplasty. Bone histomorphometry revealed a trend of low mineralized volume in T2D (T2D 0.249% [0.156–0.366]) vs non-diabetic subjects 0.352% [0.269–0.454]; p=0.053, as well as reduced bone strength (T2D 21.60 MPa [13.46–30.10] vs non-diabetic subjects 76.24 MPa [26.81–132.9]; p=0.002). We also showed that gene expression of Wnt agonists *LEF-1* (p=0.0136) and *WNT10B* (p=0.0302) were lower in T2D. Conversely, gene expression of *WNT5A* (p=0.0232), *SOST* (p<0.0001), and *GSK3B* (p=0.0456) were higher, while collagen (*COL1A1*) was lower in T2D (p=0.0482). AGEs content was associated with

*SOST* and *WNT5A* (r=0.9231, p<0.0001; r=0.6751, p=0.0322), but inversely correlated with *LEF-1* and *COL1A1* (r=−0.7500, p=0.0255; r=−0.9762, p=0.0004). *SOST* was associated with glycemic control and disease duration (r=0.4846, p=0.0043; r=0.7107, p=0.00174), whereas *WNT5A* and *GSK3B* were only correlated with glycemic control (r=0.5589, p=0.0037; r=0.4901, p=0.0051). Finally, Young's modulus was negatively correlated with *SOST* (r=−0.5675, p=0.0011), *AXIN2* (r=−0.5523, p=0.0042), and *SFRP5* (r=−0.4442, p=0.0437), while positively correlated with *LEF-1* (r=0.4116, p=0.0295) and *WNT10B* (r=0.6697, p=0.0001). These findings suggest that Wnt signaling and AGEs could be the main determinants of bone fragility in T2D.

## eLife assessment

This study provides **valuable** insights into understanding bone fragility in T2D patients through the use of human skeletal tissue, reinforcing previous pre-clinical studies or observational studies using serum samples that the Wnt signaling pathway may play a critical role in T2D-related bone impairment. The methods are **solid**, but a limited number of subjects and a small set of genes with lack of data in terms of cellular properties of skeletal tissue are viewed as weaknesses.

## Introduction

Type 2 diabetes (T2D) is a metabolic disease, with an increasing worldwide prevalence, characterized by chronic hyperglycemia and adverse effects on multiple organ systems, including bones (*Hofbauer et al., 2022*). Patients with T2D have an increased fracture risk, particularly at the hip, compared to individuals without diabetes. A recent meta-analysis reported that individuals with T2D have 1.27 relative risk of hip fracture compared to non-diabetic controls (*Wang et al., 2019*). Fragility fractures in patients with T2D occur at normal or even higher bone mineral density compared to healthy subjects, implying compromised bone quality in diabetes. T2D is associated with a reduced bone turnover (*Rubin and Patsch, 2016*), as shown by lower serum levels of biochemical markers of bone formation, such as procollagen type 1 amino-terminal propeptide and osteocalcin, and bone resorption, C-terminal cross-linked telopeptide in diabetic patients compared to non-diabetic individuals (*Napoli et al., 2018*; *Hygum et al., 2017*; *Starup-Linde and Vestergaard, 2016*; *Starup-Linde et al., 2016*). Accordingly, dynamic bone histomorphometry of T2D postmenopausal women showed a lower bone formation rate, mineralizing surface, osteoid surface, and osteoblast surface (*Manavalan et al., 2012*). Our group recently demonstrated that T2D is also associated with increased *SOST* and decreased *RUNX2* genes expression, compared to non-diabetic subjects (*Piccoli et al., 2020*). Moreover, we have proved in a diabetic model that a sclerostin-resistant *Lrp5* mutation, associated with high bone mass, fully protected bone mass and strength even after prolonged hyperglycemia (*Leanza et al., 2021*). Sclerostin is a potent inhibitor of the canonical Wnt signaling pathway, a key pathway that regulates bone homeostasis (*Maeda et al., 2019*).

Diabetes and chronic hyperglycemia are also characterized by increased advanced glycation end-products (AGEs) production and deposition (*Tan et al., 2002*). AGEs may interfere with osteoblast differentiation, attachment to the bone matrix, function, and survival (*Kume et al., 2005*; *Sanguineti et al., 2008*). AGEs also alter bone collagen structure and reduce the intrinsic toughness of bone, thereby affecting bone material properties (*Piccoli et al., 2020*; *Yamamoto and Sugimoto, 2016*; *Furst et al., 2016*). In this work, we hypothesized that T2D and AGEs accumulation downregulate Wnt canonical signaling and negatively affect bone strength. Results confirmed that T2D downregulates Wnt/beta-catenin signaling and reduces collagen mRNA levels and bone strength, in association with AGEs accumulation.

## Results

### Subject characteristics

Clinical characteristics of study subjects are presented in *Table 1*. T2D and non-diabetic subjects did not differ in age, BMI, and menopausal age. As expected, fasting glucose was significantly higher in T2D compared to non-diabetic subjects (112.00 mg/dl [104.0–130.0]) mg/dl, vs. 94.00 [87.2–106.3],

**eLife digest** Type 2 diabetes is a long-term metabolic disease characterised by chronic high blood sugar levels. This in turn has a negative impact on the health of other tissues and organs, including bones. Type 2 diabetes patients have an increased risk of fracturing bones compared to non-diabetics. This is particularly true for fragility fractures, which are fractures caused by falls from a short height (i.e., standing height or less), often affecting hips or wrists. Usually, a lower bone density is associated with higher risk of fractures. However, patients with type 2 diabetes have increased bone fragility despite normal or higher bone density.

One reason for this could be the chronically high levels of blood sugar in type 2 diabetes, which alter the properties of proteins in the body. It has been shown that the excess sugar molecules effectively 'react' with many different proteins, producing harmful compounds in the process, called Advanced Glycation End-products, or AGEs. AGEs are – in turn –thought to affect the structure of collagen proteins, which help hold our tissues together and decrease bone strength. However, the signalling pathways underlying this process are still unclear.

To find out more, Leanza et al. studied a signalling molecule, called sclerostin, which inhibits a signalling pathway that regulates bone formation, known as Wnt signaling. The researchers compared bone samples from both diabetic and non-diabetic patients, who had undergone hip replacement surgery. Analyses of the samples, using a technique called real-time-PCR, revealed that gene expression of sclerostin was increased in samples of type 2 diabetes patients, which led to a downregulation of Wnt signaling related genes. Moreover, the downregulation of Wnt genes was correlated with lower bone strength (which was measured by compressing the bone tissue). Further biochemical analysis of the samples revealed that higher sclerostin activity was also associated with higher levels of AGEs.

These results provide a clearer understanding of the biological mechanisms behind compromised bone strength in diabetes. In the future, Leanza et al. hope that this knowledge will help us develop treatments to reduce the risk of bone complications for type 2 diabetes patients.

respectively; [p=0.009]. Median hemoglobin A1c (HbA1c) was determined in all T2D subjects within 3 months before surgery (6.95% [6.37–7.37]). Median disease duration in T2D subjects was (14.50 years [7.25–19.25]). Diabetes medications included monotherapy with metformin (n=12) and combination therapy with metformin plus insulin and glinide (n=3). There were no differences in serum calcium, eGFR (CKD-EPI equation), and serum blood urea nitrogen.

## Bone histomorphometry
Bone samples of nine T2D and nine non-diabetic subjects were used for histomorphometry analysis. We found no significant differences in BV/TV and osteoid volume, while mineralized volume/total volume (Md.V/TV) trended lower in T2D subjects relative to controls (0.249% [0.156–0.336] vs 0.352% [0.269–0.454]; p=0.053) (*Table 2*).

## Bone compression tests
Young's modulus was lower in T2D compared to non-diabetic subjects (21.6 MPa [13.46–30.10] vs. 76.24 MPa [26.81–132.9]; p=0.0025), while ultimate strength and yield strength were not different between the two groups (*Table 3*).

## Gene expression
*SOST* mRNA was significantly higher in T2D than in non-diabetic subjects (*Figure 1A*, p<0.0001), whereas there was no difference in *DKK1* gene expression between the two groups (*Figure 1B*). Of note, *SOST* mRNA transcript was very low in the majority of non-diabetic subjects (*Figure 1A*). *LEF-1* (*Figure 1C*, p=0.0136), *WNT10B* (*Figure 1D*, p=0.0302), and *COL1A1* (*Figure 1F*, p=0.0482) mRNA transcripts were significantly lower in T2D compared to non-diabetic subjects. Conversely, *WNT5A* was higher in T2D relative to non-diabetics (*Figure 1E*, p=0.0232). Moreover, *GSK3B* was significantly increased in T2D compared to non-diabetic subjects (*Figure 1G*, p=0.0456), but we did not find

**Table 1.** Clinical features of the study subjects.
Results were analyzed using unpaired t-test with Welch's correction and are presented as median and percentiles (25th and 75th).

| | T2D subjects (n=15) | Non-diabetic subjects (n=21) | p-Value |
|---|---|---|---|
| Age (years) | 73.00 (67.00–80.00) | 73.00 (68.50–79.00) | 0.644 |
| BMI (kg/m$^2$) | 30.81 (24.44–34.00) | 25.00 (24.00–31.50) | 0.117 |
| Menopausal age (years) | 50.00 (42.50–52.75) | 52.00 (48.00–53.00) | 0.344 |
| Fasting glucose levels (mg/dl) | 112.00 (104.00–130.0) | 94.00 (87.25–106.3) | **0.009 |
| Disease duration (years) | 14.50 (7.25–19.25) | – | – |
| HbA1c (%) | 6.95 (6.37–7.37) | – | – |
| Serum calcium (mg/dl) | 9.05 (8.800–9.550) | 9.15 (9.000–9.550) | 0.535 |
| eGFR (ml/min/1.73 m$^2$) | 78,30 (59.90–91.10) | 75.60 (61.35–88.55) | 0.356 |
| Serum blood urea nitrogen (mg/dl) | 42.00 (36.00–53.00) | 37.00 (31.75–46.50) | 0.235 |

** p value ≤ 0.01.

any significant difference in gene expression of *AXIN2*, *BETA-CATENIN,* and *SFRP5* (*Figure 1H–J*) between our groups.

## Correlation analysis of Wnt target genes, AGEs, and glycemic control

As shown in *Figure 2*, AGEs were inversely correlated with *LEF-1* (*Figure 2A*, p=0.0255) and *COL1A1* mRNA abundance (*Figure 2B*, p=0.0004), whereas they were positively correlated with *SOST* (*Figure 2C*, p<0.0001) and *WNT5A* mRNA (*Figure 2D*, p=0.0322). There was no correlation between AGEs content and *WNT10B* (*Figure 2E*; p=0.1938) or *DKK1* gene expression (*Figure 2F*; p=0.9349). Likewise, we did not find any significant correlation between *LEF-1*, *WNT5A*, *WNT10B*, *DKK-1*, *COL1A1* expression in bone and glycemic control in T2D individuals (*Figure 3—figure supplement 1A–D*). However, there were positive correlations between *SOST* and fasting glucose levels (*Figure 3A*, p=0.0043), *SOST* and disease duration (*Figure 3B*, p=0.00174), *WNT5A*, *GSK3B*, and fasting glucose levels (*Figure 3C*, p=0.0037; *Figure 3D*, p=0.0051).

## Correlation analysis of Wnt target genes and bone mechanical parameters

As shown in *Figure 4*, Young's modulus was negatively correlated with *SOST* (*Figure 4A*, p=0.0011), *AXIN2* (*Figure 4D*, p=0.0042), and *SFRP5* (*Figure 4F*, p=0.0437), while positively correlated with *LEF-1* (*Figure 4B*, p=0.0295) and *WNT10B* (*Figure 4C*, p=0.0001). Ultimate strength was associated with *WNT10B* (*Figure 4F*, p=0.0054) and negatively correlated with *AXIN2* (*Figure 4G*, p=0.0472). Finally, yield strength was associated with *LEF-1* (*Figure 4H*, p=0.0495) and *WNT10B* (*Figure 4I*, p=0.0020) and negatively correlated with *GSK3B* (*Figure 4J*, p=0.0245), AXIN2 (*Figure 4K*, p=0.0319), and *SFRP5* (*Figure 4L*, p=0.0422). Non-significant correlations are reported in *Figure 4—figure supplement 1A–Q*.

## Discussion

We show that key components of the Wnt/beta-catenin signaling are abnormally expressed in the bone of postmenopausal women with T2D and they are associated with AGEs and reduced bone strength (*Figure 5*). *LEF-1*, a transcription factor that mediates responses to Wnt signal and Wnt target genes itself, and *WNT10B*, an endogenous regulator of Wnt/beta-catenin signaling and skeletal progenitor cell fate, are both downregulated in bone of postmenopausal women with T2D. Consistently, in this group, the expression of the Wnt inhibitor, *SOST* is increased, suggesting suppression

**Table 2.** Histomorphometric parameters of trabecular bone of the study subjects.
Results were analyzed using unpaired t-test with Welch's correction and are presented as median and percentiles (25th and 75th).

| | T2D subjects (n=9) | Non-diabetic subjects (n=9) | p-Value |
|---|---|---|---|
| BV/TV (%) | 0.248 (0.157–0.407) | 0.358 (0.271–0.456) | 0.120 |
| Md.V/BV (%) | 0.994 (0.984–0.998) | 0.995 (0.985–0.997) | 0.998 |
| Md.V/TV (%) | 0.249 (0.156–0.366) | 0.352 (0.269–0.454) | 0.053 |
| OV/BV (%) | 0.009 (0.002–0.009) | 0.004 (0.002–0.015) | 0.704 |
| OV/TV (%) | 0.001 (0.0002–0.0058) | 0.001 (0.0007–0.0056) | 0.896 |
| OS/BS (%) | 0.026 (0.022–0.161) | 0.035 (0.009–0.117) | 0.525 |

of Wnt/beta-catenin signaling. Interestingly, our data suggest that sclerostin expression is very low in healthy postmenopausal women not affected by osteoporosis. Moreover, we reported an increase in the expression level of bone *GSK3B,* in line with downregulated Wnt/beta-catenin signaling in T2D. Our data also show that the expression of *WNT5A,* a non-canonical ligand linked to inhibition of Wnt/beta-catenin signaling, was increased, whereas *COL1A1* was decreased. These findings are consistent with reduced bone formation and suppression of Wnt signaling in T2D. We have previously reported upregulation of *SOST* and downregulation of *RUNX2* mRNA in another cohort of postmenopausal women with T2D (*Piccoli et al., 2020*). Of note, the cohort of T2D subjects studied here had glycated hemoglobin within therapeutic targets, implying that the changes in gene transcription we identified persist in T2D bone despite good glycemic control.

High circulating sclerostin has been reported in diabetes (*García-Martín et al., 2012*; *Gennari et al., 2012*), and increased sclerostin is associated with fragility fractures (*Yamamoto et al., 2013*). Aside from confirming higher *SOST* expression, we also show that other Wnt/beta-catenin osteogenic ligands are abnormally regulated in the bone of T2D postmenopausal women. *WNT10B* is a positive regulator of bone mass; transgenic overexpression in mice results in increased bone mass and strength (*Longo et al., 2004*), whereas genetic ablation of *WNT10B* is characterized by reduced bone mass (*Bennett et al., 2005*; *Kubota et al., 2009*), and decreased number and function of osteoblasts (*Bennett et al., 2005*). More to the point, *WNT10B* expression is reduced in the bone of diabetic mice (*Zhang et al., 2015*). Therefore, the reduced *WNT10B* in human bone we found in the present study further supports the hypothesis of reduced bone formation in T2D. Accordingly, *LEF-1* gene expression was also downregulated confirming that Wnt/beta-catenin pathway is decreased in T2D. Importantly, the overexpression of *LEF-1* induces the expression of osteoblast differentiation genes (osteocalcin and *COL1A1*) in differentiating osteoblasts (*Hoeppner et al., 2009*). In fact, in this study we also demonstrated that a downregulation of *LEF-1* in T2D bone goes along with a downregulation of *COL1A1*, strengthen data of a reduced production of bone matrix most likely as the result of reduced osteoblasts synthetic activity in diabetes (*Manavalan et al., 2012*; *Khan et al., 2015*). Reduced *RUNX2* in T2D postmenopausal women also confirms previous

**Table 3.** Bone mechanical parameters of trabecular bone of the study subjects.
Results were analyzed using unpaired t-test with Welch's correction and are presented as median and percentiles (25th and 75th).

| | T2D subjects (n=11) | Non-diabetic subjects (n=21) | p-Value |
|---|---|---|---|
| Young's modulus (MPa) | 21.60 (13.46–30.10) | 76.24 (26.81–132.9) | 0.002 |
| Ultimate strength (MPa) | 3.015 (2.150–13.86) | 7.240 (3.150–8.898) | 0.914 |
| Yield strength (MPa) | 2.525 (1.943–6.393) | 6.150 (3.115–7.423) | 0.159 |

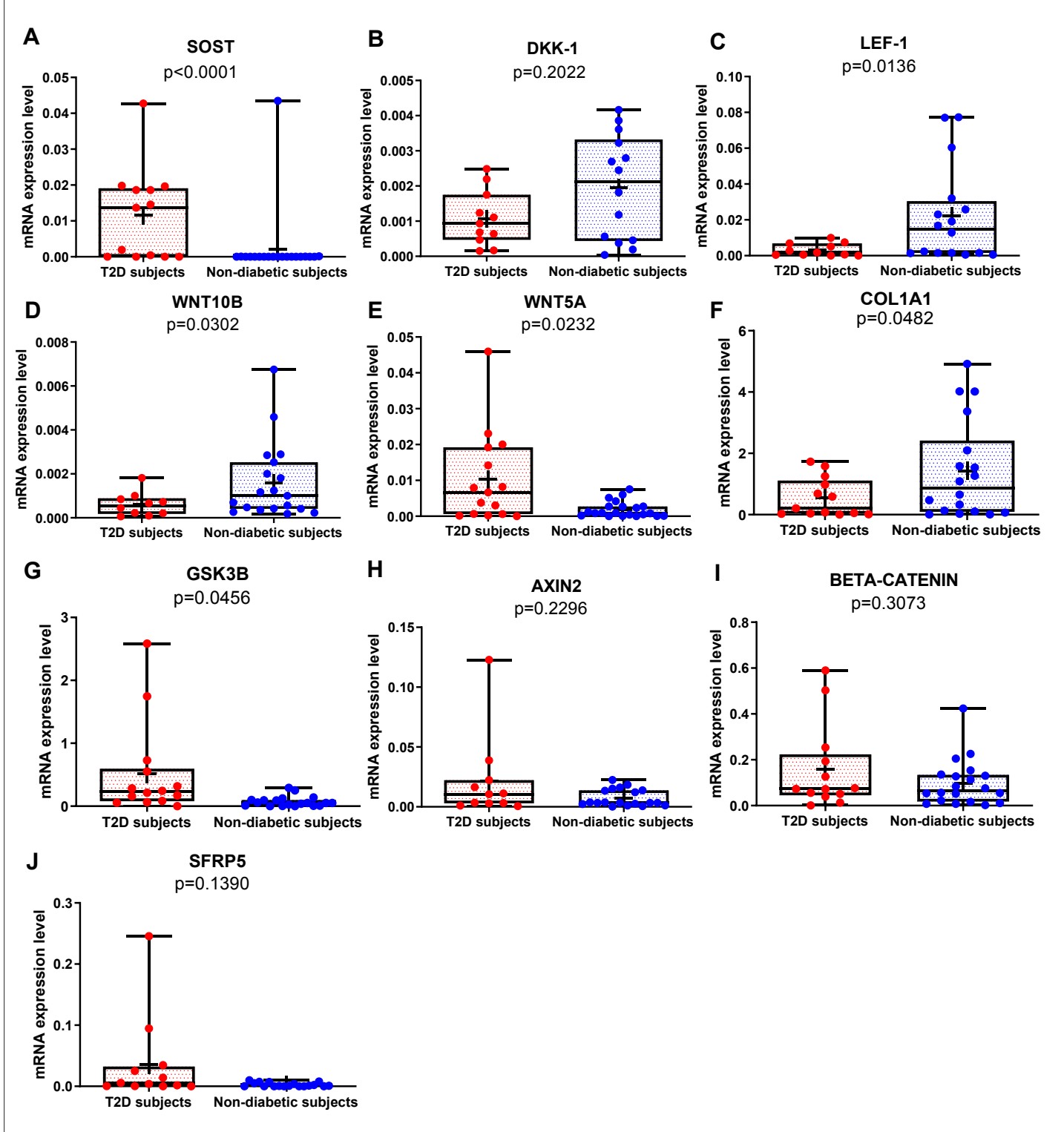

**Figure 1.** Gene expression analysis in trabecular bone samples. (**A**) SOST mRNA levels resulted higher in type 2 diabetes (T2D) subjects versus non-diabetic subjects (p<0.0001). (**B**) DKK-1 mRNA expression level was not different between groups (p=0.2022). (**C**) LEF-1 mRNA levels resulted lower in T2D subjects versus non-diabetics subjects (p=0.0136). (**D**) WNT10B mRNA expression level was lower in T2D subjects versus non-diabetic subjects (p=0.0302). (**E**) WNT5A mRNA resulted higher in T2D subjects versus non-diabetics subjects (p=0.0232). (**F**) COL1A1 mRNA levels resulted lower in T2D subjects versus non-diabetic subjects (p=0.0482). (**G**) GSK3B mRNA levels resulted higher in T2D subjects versus non-diabetic subjects (p=0.0456). (**H–**

*Figure 1 continued on next page*

*Figure 1 continued*

J) AXIN2, BETA-CATENIN, SFRP5 mRNA levels were not different between groups (p=0.2296, p=0.3073, p=0.1390). Data are expressed as fold changes over beta-actin. Medians and interquartile ranges, differences between non-diabetics and T2D subjects were analyzed using Mann-Whitney test.

The online version of this article includes the following source data for figure 1:

**Source data 1.** Data represented by each point in *Figure 1A–J*.

findings (*Piccoli et al., 2020*) and further supports the notion of reduced osteoblast differentiation or function in diabetes. On the other hand, the contribution of upregulated *WNT5A* in diabetic bone is more complex. *WNT5A* regulates Wnt/beta-catenin signaling depending on the receptor availability (*Mikels and Nusse, 2006*). Non-canonical *WNT5A* activates beta-catenin-independent signaling, including the Wnt/Ca++ (*Dejmek et al., 2006*) and planar cell polarity pathways (*Oishi et al., 2003*). Heterozygous Wnt5a null mice have low bone mass with impaired osteoblast and osteoclast differentiation (*Maeda et al., 2012*). Wnt5a inhibits Wnt3a protein by downregulating beta-catenin-induced reporter gene expression (*Mikels and Nusse, 2006*). In line with these findings, we showed that there was an increased gene expression of *WNT5A* in bone of T2D postmenopausal women, confirming a downregulated Wnt/beta-catenin signaling and impaired osteoblasts function. Moreover, *GSK3B* is a widely expressed serine/threonine kinase involved in multiple pathways regulating immune cell activation and glucose metabolism. Preclinical studies reported that *GSK3B* is a negative regulator of Wnt/beta-catenin signaling and bone metabolism (*McManus et al., 2005*; *Chen et al., 2021*), and its increase is associated with T2D and alterations in insulin secretion and sensitivity (*Nunez Lopez et al., 2022*; *Xia et al., 2022*). Our data confirmed that *GSK3B* is increased in T2D postmenopausal women and it is associated with reduced yield strength. In fact, we also showed an impaired bone mechanical plasticity in T2D, in line with other studies showing a reduced bone strength (*Furst et al., 2016*; *Farr et al., 2014*; *Hunt et al., 2019*). In addition, this study reported significant correlations of bone mechanical parameters and Wnt target genes, which might reflect the biological effect of downregulated Wnt signaling and AGEs accumulation on bone mechanical properties in diabetes.

We have previously shown that AGEs content is higher in T2D bone compared to non-diabetic bone, even in patients with well-controlled T2D (*Piccoli et al., 2020*). Here, we show that AGEs accumulation is positively correlated with *SOST*, *WNT5A,* and *GSK3B* gene expression, and negatively correlated with *LEF-1*, *WNT10B*, and *COL1A1* mRNA. These findings are consistent with the hypothesis that AGEs accumulation is associated with impaired Wnt signaling and low bone turnover in T2D. We did not find any abnormalities in histomorphometric parameters in our subjects with T2D, consistent with our previous report (*Piccoli et al., 2020*). Reduced osteoid thickness and osteoblast number were reported in premenopausal T2D women with poor glycemic control compared to non-diabetic subjects but not in the group with good glycemic control (*Andrade et al., 2020*). Therefore, good glycemic control appears to prevent or rescue any changes in static histologic parameters of bone turnover that might be caused by uncontrolled diabetes.

Our study has some limitations. One is the cross-sectional design; another one is the relatively small number of T2D subjects enrolled. Moreover, we measured the mRNA abundance of the genes of interest, and we cannot assume that the differences we found reflect differences in protein abundance. Although osteoarthritis may affect some of the genes we studied (*Weivoda et al., 2017*), all study subjects were affected by variable degree of osteoarthritis, and the effect of such potential confounder is not likely to be different between T2D and control subjects. Finally, we did not use the tetracycline double-labeled technique to investigate dynamic bone parameters.

The main strength of our study is that this study is the first to explore the association of AGEs on Wnt pathway in postmenopausal T2D women. Moreover, we measured the expression of several Wnt genes directly on bone samples of postmenopausal T2D women.

In conclusion, our data show that, despite good glycemic control, T2D decreases expression of *COL1A1* and Wnt genes that regulate bone turnover, in association with increased AGEs content and reduced bone strength. These results may help understand the mechanisms underlying bone fragility in T2D.

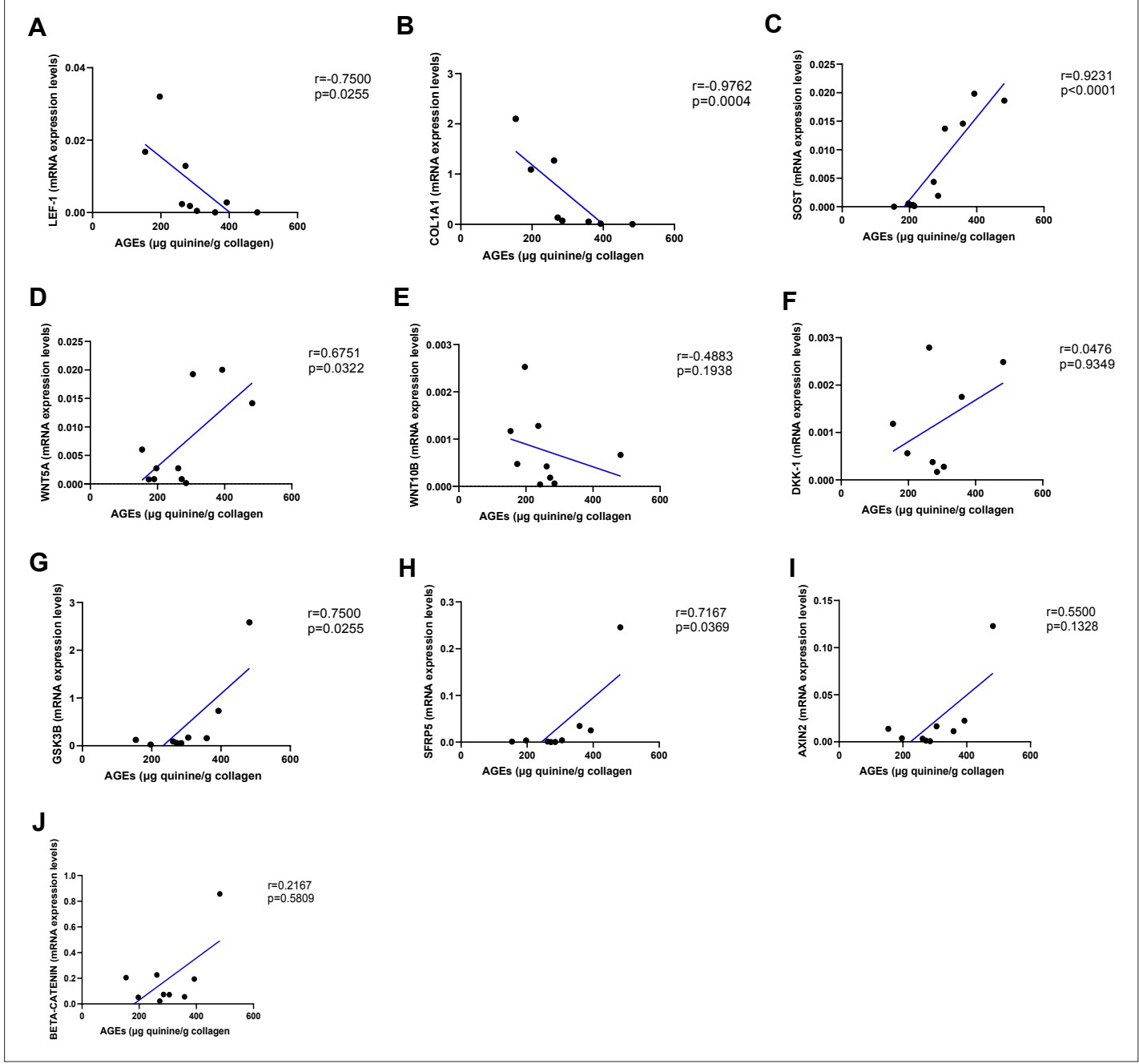

**Figure 2.** Relationship between advanced glycation end-products (AGEs) (μg quinine/g collagen) bone content and mRNA level of the Wnt signaling key genes in type 2 diabetes (T2D) and non-diabetic subjects. (**A**) LEF-1 negatively correlated with AGEs (r=−0.7500; p=0.0255). (**B**) COL1A1 negatively correlated with AGEs (r=−0.9762; p=0.0004). (**C**) SOST mRNA level expression positively correlated with AGEs (r=0.9231; p<0.0001). (**D**) WNT5A mRNA expression level positively correlated with AGEs (r=0.6751; p=0.0322). (**E**) WNT10B mRNA expression level was not correlated with AGEs (r=−0.4883; p=0.1938). (**F**) DKK1 mRNA expression level was not correlated with AGEs (r=0.0476; p=0.9349). (**G**) GSK3B mRNA expression level was positively correlated with AGEs (r=0.7500; p=0.0255). (**H**) SFRP5 mRNA expression level was positively correlated with AGEs (r=0.7167; p=0.0369). (**I**) AXIN2 and (**J**) SFRP5 mRNA expression levels were not correlated with AGEs (r=0.5500, p=0.1328; r=0.2167, p=0.5809). Data were analyzed using nonparametric Spearman correlation analysis and r represents the correlation coefficient.

The online version of this article includes the following source data for figure 2:

**Source data 1.** Data represented by each point in *Figure 2A–J*.

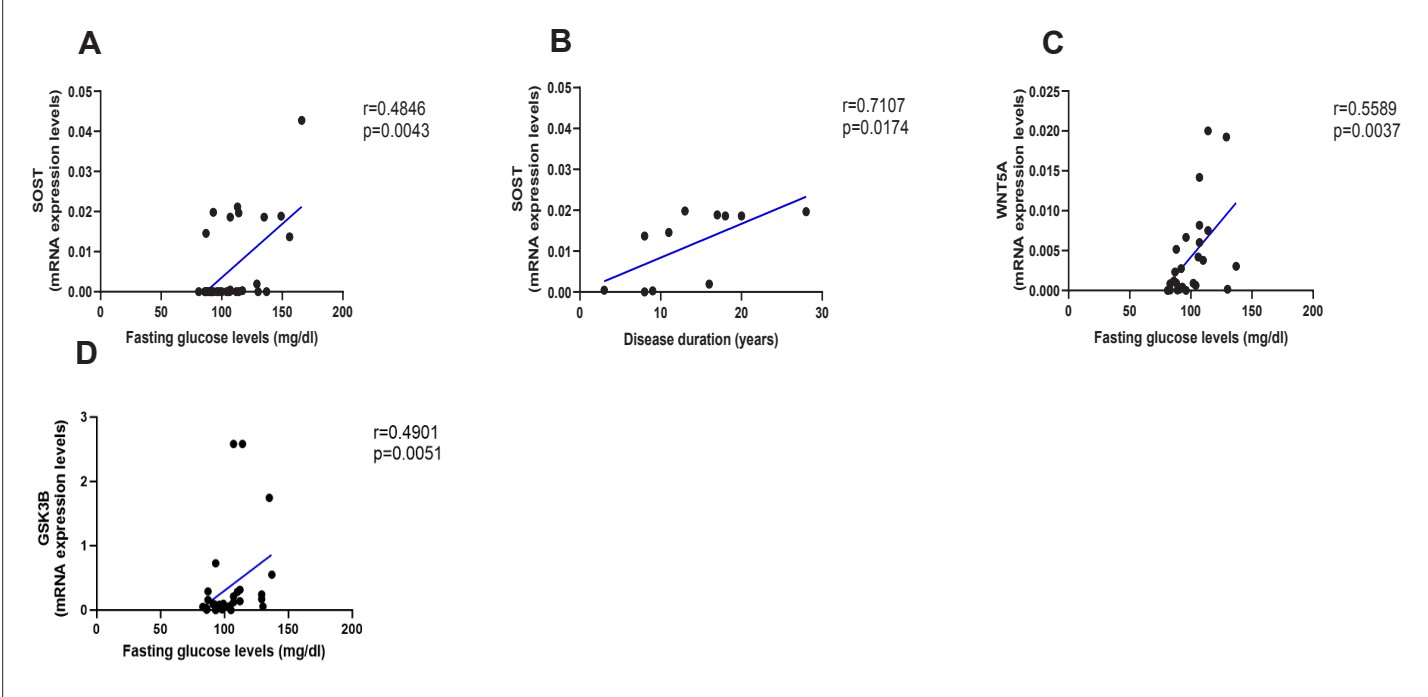

**Figure 3.** Relationship between fasting glucose levels (mg/dl) and disease duration with SOST and WNT5A mRNA levels. (**A**) SOST positively correlated with fasting glucose levels (r=0.4846; p=0.0043). (**B**) SOST positively correlated with disease duration (r=0.7107; p=0.0174). (**C**) WNT5A positively correlated with fasting glucose levels (r=0.5589; p=0.0037). (**D**) GSK3B positively correlated with fasting glucose levels (r=0.4901; p=0.0051). Data were analyzed using nonparametric Spearman correlation analysis and r represents the correlation coefficient.

The online version of this article includes the following source data and figure supplement(s) for figure 3:

**Source data 1.** Data represented by each point in *Figure 3A–D*.

**Figure supplement 1.** Relationship between fasting glucose levels (mg/dl) and LEF 1, WNT5A, WNT10B, DKK-1, COL1A1 mRNA levels.

## Materials and methods
### Study subjects

We enrolled a total of 36 postmenopausal women (15 with T2D and 21 non-diabetic controls) under-going hip arthroplasty for osteoarthritis, consecutively screened to participate in this study between 2020 and 2022. Diabetes status was confirmed by the treating diabetes physician. Participants were diagnosed with diabetes when they had fasting plasma glucose ≥126 mg/dl or 2 hr plasma glucose≥200 mg/dl during a 75 g oral glucose tolerance test; or HbA1c≥6.5% in accordance with the American Diabetes Association diagnostic criteria. Eligible participants were ≥60 years of age. Exclusion criteria were any diseases affecting bone or malignancy. Additionally, individuals treated with medications affecting bone metabolism such as estrogen, raloxifene, tamoxifen, bisphospho-nates, teriparatide, denosumab, thiazolidinediones, glucocorticoids, anabolic steroids, and phenytoin, and those with hypercalcemia or hypocalcemia, hepatic or renal disorder, hypercortisolism, current alcohol, or tobacco use were excluded. The study was approved by the Ethics Committee of Campus Bio-Medico University of Rome (Prot..42/14 PT_ComEt CBM) and all participants provided written informed consent. All procedures were conducted in accordance with the Declaration of Helsinki.

### Specimen preparation

Femoral head specimens were obtained during hip arthroplasty. As described previously (*Piccoli et al., 2020*), trabecular bone specimens were collected fresh and washed multiple times in sterile PBS until the supernatant was clear of blood. Bone samples were stored at –80°C until analysis.

### Bone histomorphometry

Trabecular bone from femur heads was fixed in 10% neutral buffered formalin for 24 hr prior to storage in 70% ethanol. Tissues were embedded in methylmethacrylate and sectioned sagittally by

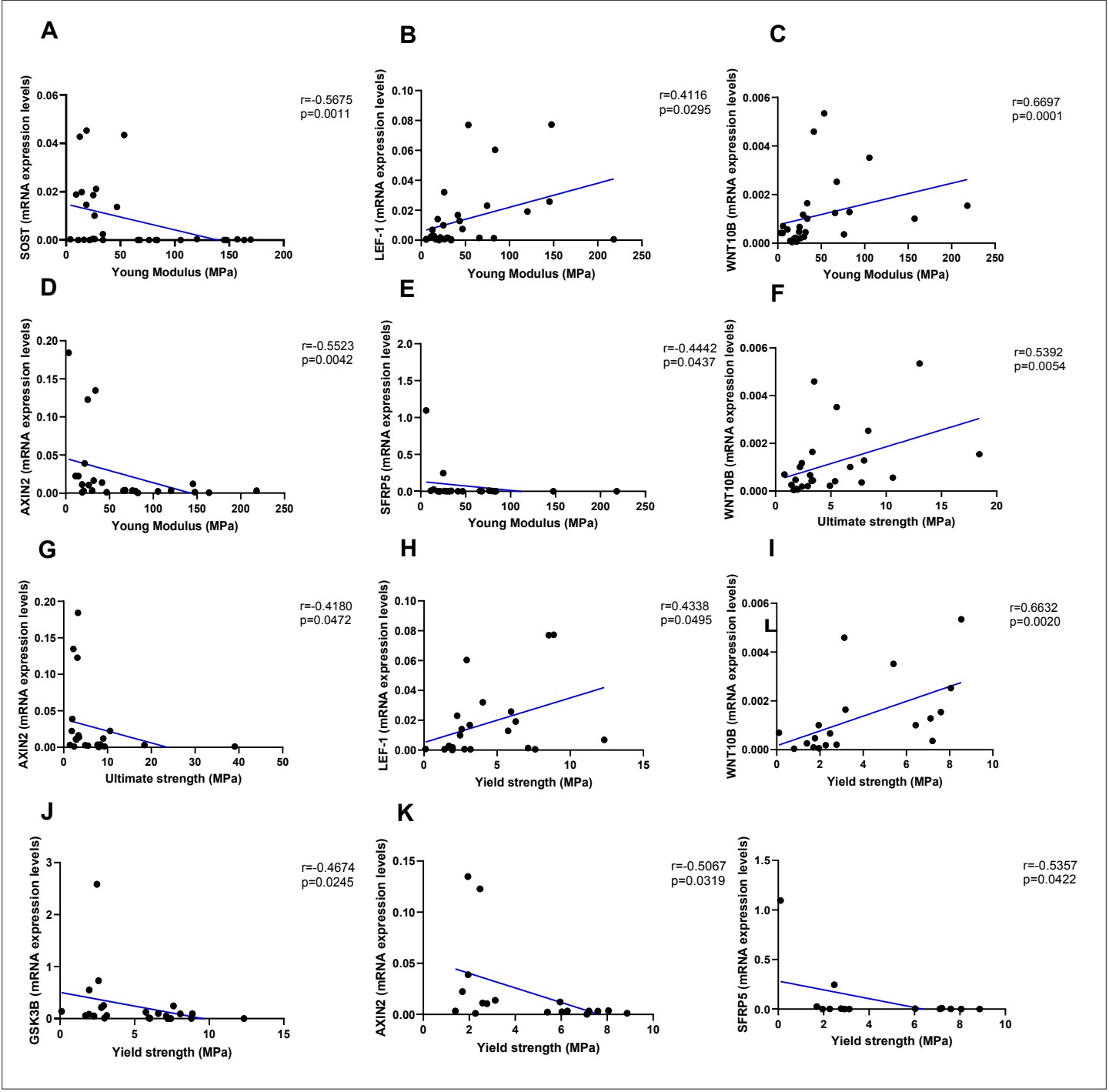

**Figure 4.** Relationship between Young's modulus (MPa), ultimate strength (MPa), and yield strength (MPa) with mRNA levels of the Wnt signaling key genes in type 2 diabetes (T2D) and non-diabetic subjects. (**A**) SOST negatively correlated with Young's modulus (MPa); (r=−0.5675; p=0.0011). (**B**) LEF-1 positively correlated with Young's modulus (MPa); (r=0.4116; p=0.0295). (**C**) WNT10B positively correlated with Young's modulus (MPa); (r=0.6697; p=0.0001). (**D**) AXIN2 negatively correlated with Young's modulus (MPa); (r=−0.5523; p=0.0042). (**E**) BETA-CATENIN negatively correlated with Young's modulus (MPa); (r=−0.5244; p=0.0050). (**F**) SFRP5 negatively correlated with Young's modulus (MPa); (r=−0.4442; p=0.0437). (**G**) WNT10B positively correlated with ultimate strength (MPa); (r=0.5392; p=0.0054). (**H**) AXIN2 negatively correlated with ultimate strength (MPa); (r=−0.4180; p=0.0472). (**I**) BETA-CATENIN negatively correlated with ultimate strength (MPa); (r=−0.5528; p=0.0034). (**J**) LEF-1 positively correlated with yield strength (MPa); (r=0.4338; p=0.0495). (**K**) WNT10B positively correlated with yield strength (MPa); (r=0.6632; p=0.0020). (**L**) GSK3B negatively correlated with yield strength (MPa); (r=−0.4674; p=0.0245). (**M**) AXIN2 negatively correlated with yield strength (MPa); (r=−0.5067; p=0.0319). (**N**) BETA-CATENIN negatively

*Figure 4 continued on next page*

*Figure 4 continued*

correlated with yield strength (MPa); (r=−0.5491; p=0.0149). (**O**) SFRP5 negatively correlated with yield strength (MPa); (r=−0.5357; p=0.0422). Data were analyzed using nonparametric Spearman correlation analysis and r represents the correlation coefficient.

The online version of this article includes the following source data and figure supplement(s) for figure 4:

**Source data 1.** Data represented by each point in *Figure 4A–L*.

**Figure supplement 1.** Relationship between Young's modulus (MPa), ultimate strength (MPa), and yield strength (MPa) with mRNA levels of the Wnt signaling genes in type 2 diabetes (T2D) and non-diabetic subjects.

the Washington University Musculoskeletal Histology and Morphometry Core. Sections were stained with Goldner's trichrome. Then, a rectangular region of interest (ROI) containing trabecular bone was chosen below the cartilage-lined joint surface and primary spongiosa. This region had an average dimension of 45 mm². Tissue processing artifacts, such as folding and edges, were excluded from the ROI. A threshold was chosen using the Bioquant Osteo software to automatically select trabeculae and measure bone volume. Finally, Osteoid was highlighted in the software and quantified semi-automatically using a threshold and correcting with the brush tool. Unstained and TRAP-stained (Sigma) slides were imaged at ×20 high resolution using a NanoZoomer 2.0 with bright field and FITC/TRITC (Hamamatsu Photonics). Images were then analyzed via Bioquant Osteo software according to the manufacturer's instructions and published standards (v18.2.6, Bioquant Image Analysis Corp., Nashville, TN, USA).

## Bone compression tests

We used cylindrical bone specimens of trabecular core (with a diameter of 10 mm and a length of 20 mm) from 11 T2D and 21 non-diabetic subjects to measure bone mechanical parameters (Young's modulus, ultimate strength, and yield strength), as previously described (*Piccoli et al., 2020*).

## RNA extraction and gene expression by RT-PCR

Total RNA from trabecular bone samples was extracted using TRIzol (Invitrogen) following the manufacturer's instructions. The concentration and purity of the extracted RNA were assessed spectrophotometrically (TECAN, InfiniteM200PRO), and only samples with 260/280 absorbance ratio between

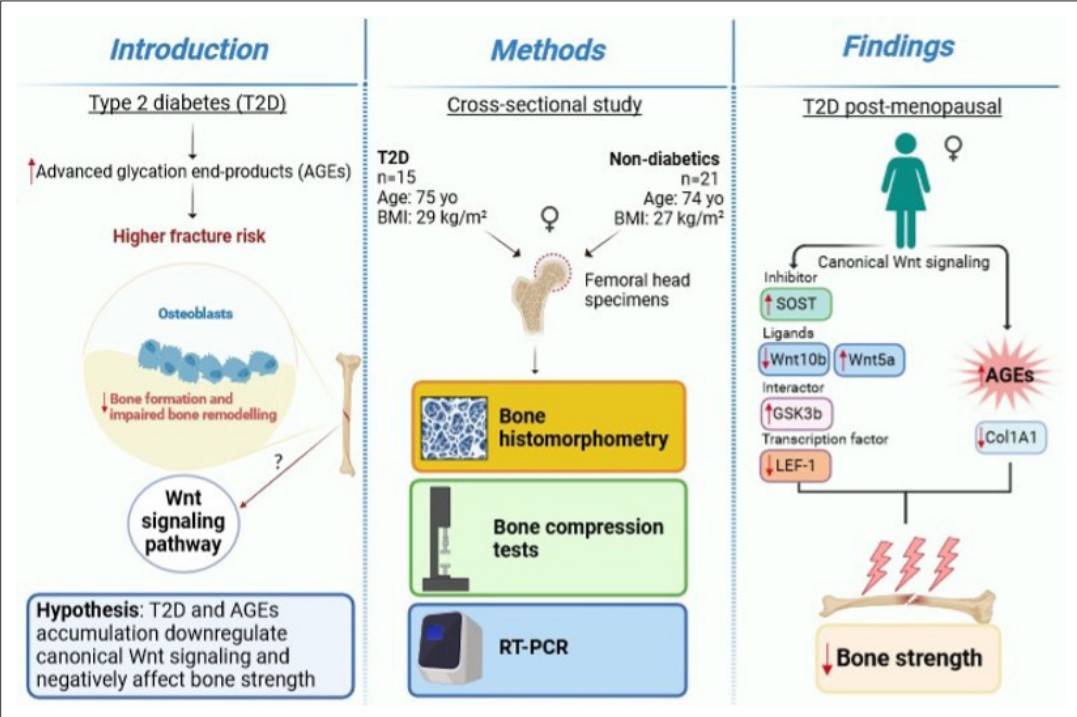

**Figure 5.** A graphical summary of the study.

1.8 and 2 were used for reverse transcription using High-Capacity cDNA Reverse Transcription Kit (Applied Biosystems, Carlsbad, CA, USA) according to the manufacturer's recommendations. Transcription products were amplified using TaqMan real-time PCR (Applied Biosystems, Carlsbad, CA, USA) and a standard protocol (95°C for 10 min; 40 cycles of 95°C for 15 s and 60°C for 1 min; followed by 95°C for 15 s, 60°C for 15 s, and 95°C for 15 s). *Beta-actin* expression was used as an internal control (housekeeping gene). Relative expression levels of Sclerostin (*SOST*), Dickkopf-1 (*DKK-1*), Wnt ligands (*WNT5A* and *WNT10B*), T-cell factor/lymphoid enhancer factor 1 (*LEF-1*), collagen type I alpha 1 chain (*COL1A1*), glycogen synthase kinase 3 beta (*GSK3B*), axis inhibition protein 2 (*AXIN2*), beta-catenin (*BETA-CATENIN*), and secreted frizzled-related protein 5 (*SFRP5*) were calculated using the $2^{-\Delta Ct}$ method.

## Statistical analysis

Data were analyzed using GraphPad Prism 9.0 (GraphPad Software, San Diego, CA, USA). Patients' characteristics were described using means and standard deviations or medians and 25th–75th percentiles, as appropriate, and percentages. Group data are presented in boxplots with median and interquartile range; whiskers represent maximum and minimum values. We assessed data for normality and Mann-Whitney test was used to compare variables between groups. Data were analyzed using nonparametric Spearman correlation analysis and the correlation coefficients (r) were used to assess the relationship between variables. We used Grubbs' test to assess and exclude outliers. For bone histomorphometry, we performed a priori sample size calculation using G*Power 3.1.9.7, based on the t-test, difference between two independent groups setting. Analysis demonstrated that given an effect size of 2.2776769 (*Manavalan et al., 2012*), we needed a total of 12 patients (6/group) to reach a power of 0.978.

## Acknowledgements

This work was supported by an internal Grant of Campus Bio-Medico University of Rome.

## Additional information

### Funding

| Funder | Grant reference number | Author |
|---|---|---|
| Università Campus Bio-Medico di Roma | Internal grant | Mauro Maccarrone<br>Rocco Papalia<br>Nicola Napoli |

The funders had no role in study design, data collection and interpretation, or the decision to submit the work for publication.

### Author contributions

Giulia Leanza, Conceptualization, Data curation, Software, Formal analysis, Supervision, Validation, Investigation, Writing - original draft, Project administration, Writing – review and editing; Francesca Cannata, Resources, Data curation, Investigation; Malak Faraj, Data curation, Investigation, Visualization, Methodology, Writing – review and editing; Claudio Pedone, Conceptualization, Data curation, Software, Formal analysis, Investigation, Writing – review and editing; Viola Viola, Data curation, Investigation, Methodology, Writing – review and editing; Flavia Tramontana, Investigation, Methodology, Writing – review and editing; Niccolò Pellegrini, Software, Formal analysis, Investigation, Visualization, Methodology, Writing – review and editing; Gianluca Vadalà, Resources, Supervision, Funding acquisition, Investigation, Methodology; Alessandra Piccoli, Data curation, Formal analysis, Investigation, Methodology; Rocky Strollo, Conceptualization, Supervision, Visualization, Writing – review and editing; Francesca Zalfa, Writing – review and editing; Alec T Beeve, Erica L Scheller, Simon Y Tang, Investigation, Visualization, Methodology, Writing – review and editing; Roberto Civitelli, Conceptualization, Visualization, Writing – review and editing; Mauro Maccarrone, Resources, Supervision, Visualization, Writing – review and editing; Rocco Papalia, Conceptualization, Resources, Supervision, Funding acquisition, Investigation, Project administration; Nicola Napoli, Conceptualization, Data

curation, Supervision, Funding acquisition, Investigation, Visualization, Project administration, Writing – review and editing

### Author ORCIDs
Giulia Leanza ⬥ http://orcid.org/0000-0001-5489-6613
Niccolò Pellegrini ⬥ http://orcid.org/0000-0003-4539-7614
Erica L Scheller ⬥ http://orcid.org/0000-0002-1551-3816
Roberto Civitelli ⬥ http://orcid.org/0000-0003-4076-4315
Nicola Napoli ⬥ https://orcid.org/0000-0002-3091-8205

### Ethics

Human subjects: The study was approved by the Ethics Committee of the Campus Bio-Medico University of Rome (Prot..42/14 PT_ComEt CBM) and all participants provided written informed consent. All procedures were conducted in accordance with the Declaration of Helsinki.

Reviewer #1 (Public review): https://doi.org/10.7554/eLife.90437.3.sa1
Reviewer #2 (Public review): https://doi.org/10.7554/eLife.90437.3.sa2
Author response https://doi.org/10.7554/eLife.90437.3.sa3

---

## Additional files

### Supplementary files
• MDAR checklist

### Data availability

All data generated or analysed during this study are included in the manuscript and supporting files; source data files have been provided for all tables and figures of the manuscript, including figure supplements.

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
