## [Editor Report · eLife assessment]

This study provides **valuable** insights into understanding bone fragility in T2D patients through the use of human skeletal tissue, reinforcing previous pre-clinical studies or observational studies using serum samples that the Wnt signaling pathway may play a critical role in T2D-related bone impairment. The methods are **solid**, but a limited number of subjects and a small set of genes with lack of data in terms of cellular properties of skeletal tissue are viewed as weaknesses.

---

## [Referee Report · Reviewer #1 (Public review)]

Summary: Leanza et al. investigated the regulation of Wnt signaling factors in the bone tissue obtained from individuals with or without type 2 diabetes. They showed that typical canonical Wnt ligands and downstream factors (Wnt10b, LEF1) are down-regulated, while Wnt5a and sclerostin mRNA is unregulated in diabetic bone tissue. Further, Wnt5a and sclerostin associated with the content of AGEs and SOST mRNA levels also correlated with glycemic control and disease duration.

Strengths:

- A strength of the study is the investigation of Wnt signaling in bone tissue from humans with type 2 diabetes. Most studies measure only serum levels of Wnt inhibitors, but this study takes it further and looks into bone specifically.

- The measurement of AGEs and its correlation to the Wnt signaling molecules is interesting and important. The correlation of sclerostin and Wnt5a with AGEs and disease duration suggests that inhibited Wnt signaling is paralleled by higher AGE levels and potentially weaker bone.

- The methodology in terms of obtaining the bone samples and the rigorous evaluation of RNA integrity is great and provides a solid basis for further analyses.

Weaknesses:

- A weakness may include the rather limited number of samples.

Overall, this study validates findings from the group that have reported similar findings in 2020. This validates their methodology and shows that alterations in Wnt signaling are reproducible in human bone tissue.

---

## [Referee Report · Reviewer #2 (Public review)]

Summary:

This study reports the levels of expression of selected genes implicated in Wnt signaling in trabecular bone from femur heads obtained after surgery from post-menopausal women with (15 women) or without (21 women) type 2 diabetes. They find higher expression levels of SOST and WNT5A, and lower expression levels of LEF-1 and WNT10B in tissues from subjects with T2D, correlating with glycemia and advanced glycation products. No significant differences in bone density were observed. Overall, this is a cross-sectional, observational study measuring a limited set of genes found to vary with glycemia in postmenopausal women undergoing hip surgery.

Strengths:

The study demonstrates the feasibility of measuring gene expression in post-surgical trabecular bon samples and finds differences associated with glycemia despite a relatively small number of subjects. It can form the basis for further research on the causes and consequences of changes in elements of the WNT signaling pathway in bone biology and disease.

Weaknesses:

The small number of targeted genes does not provide a comprehensive view of the transcriptional landscape within which the effects are observed. The gene expression changes are not associated with cellular or physiological properties of the tissue, raising questions about the biological significance of the observations.

---

## [Author Response]

The following is the authors’ response to the original reviews.

**REVIEWER #1**
Leanza et al. investigated the regulation of Wnt signaling factors in the bone tissue obtained from individuals with or without type 2 diabetes. They showed that typical canonical Wnt ligands and downstream factors (Wnt10b, LEF1) are down-regulated, while Wnt5a and sclerostin mRNA are unregulated in diabetic bone tissue. Further, Wnt5a and sclerostin associated with the content of AGEs and SOST mRNA levels also correlated with glycemic control and disease duration.Strengths:A strength of the study is the investigation of Wnt signaling in bone tissue from humans with type 2 diabetes. Most studies measure only serum levels of Wnt inhibitors, but this study takes it further and looks into bone specifically.The measurement of AGEs and its correlation to the Wnt signaling molecules is interesting and important. The correlation of sclerostin and Wnt5a with AGEs and disease duration suggests that inhibited Wnt signaling is paralleled by higher AGE levels and potentially weaker bone.The methodology in terms of obtaining the bone samples and the rigorous evaluation of RNA integrity is great and provides a solid basis for further analyses.Weaknesses:A weakness may include the rather limited number of samples. Especially for some sub-analyses (e.g. RNA analyses), only a subset of samples was used.How was the sample size determined? It seems like more samples might have been necessary to obtain significant results for methods with a higher standard deviation (e.g. histomorphometry).

We apology for the oversight in the description of the statistical analysis and we thank the reviewer for the careful reading. For sample size calculation of bone histomorphometry we used the cohort of the only paper analyzing trabecular bone in T2D postmenopausal women by dynamic histomorphometry (Manavalan JS et al, JCEM 2012). We performed a priori sample size calculation using G*Power 3.1.9.7., based on the t-test, difference between two independent groups setting. Analysis demonstrated that given an effect size of 2.2776769, we needed a total of 12 patients (6/group) to reach a power of 0.978. Regarding gene expression analyses, it was performed not in a subset of patients, but in all recruited subjects for this study. Based on the results of gene expression analysis on our main outcome (Wnt signaling), we demonstrated that for SOST gene the effect size was 1.2733824, with a power of 0.9490065, confirming that sample size was sufficient to achieve adequate statistical power.

Why is the number of samples different for the mRNA measurements? In most cases, there were 9, but in some 8 and in some 10?

We sincerely thank the reviewer for the opportunity to clarify such important aspects. The number of samples used for mRNA quantification may differ between the different analyzed genes due to multiple reasons: First, we used for the real-time PCR only samples with high quality ratio (260/280) between 1.8-2.0 as stated in the method section of the manuscript (Page 8, lines 163-164). Moreover, we decided not to use the undetermined values, undetectable after the amplification cycles (40 cycles in total), as specified in the method section (Page 8, line 167).

Overall, this study validates findings from the group that reported similar findings in 2020. This validates their methodology and shows that alterations in Wnt signaling are reproducible in human bone tissue.

We thank the reviewer for the positive comment, we really value her/his opinion.

COMMENTS:(1) The authors could provide more details on how much of the bone was analyzed for bone histomorphometry (what area?).

We truly thank the reviewer for allowing us to explain more in depth our methodology. First, a biopsy containing trabecular bone from the femoral head was fixed in 10% neutral buffered formalin for 24 h prior to storage in 70% ethanol. Tissues were embedded in methylmethacrylate and sectioned sagittally by the Washington University Musculoskeletal Histology and Morphometry Core. Sections were stained with Goldner’s trichrome. Then, a rectangular region of interest containing trabecular bone was chosen below the cartilage-lined joint surface and primary spongiosa. This region had an average dimension of 45 mm2. Tissue processing artifacts, such as folding and edges, were excluded from the ROI. A threshold was chosen using the BIOQUANT software to automatically select trabeculae and measure bone volume. Finally, Osteoid was highlighted in the software and quantified semi-automatically using a threshold and correcting with the brush tool (as shown in the image below).

We specify that in the methods section (Page 7, lines 146-152).

(2) Could the number of samples used for histomorphometry be increased? That may also lead to more significant results.

We sincerely appreciated this suggestion from the reviewer but unfortunately, all available samples for histomorphometry have been analyzed and we are not able to increase the number of recruited participants at this time. Recruitment of people with T2D undergoing hip replacement is extremely difficult giving the limited number of those approved for elective surgery and compliant with our inclusion criteria. Considering also the long time needed to process bone sample for gene expression and histology analysis would require several months to have a consistent increase in recruited subjects. However, we have previously calculated sample size for bone histomorphometry analysis using the only available data of trabecular bone in T2D postmenopausal women measured by dynamic histomorphometry (Manavalan JS et al, JCEM 2012). We performed a priori sample size calculation using G*Power 3.1.9.7., based on the t-test of two independent groups. Analysis demonstrated that given an effect size of 2.2776769, we needed a total of 12 patients (6/group) to reach a power of 0.978.

(3) It would have been interesting to assess the biomechanical behavior of the bone specimens. While it is known that BMD is often higher in patients with T2D, the resistance to fractures is lower. Ideally, bone strength measures could be correlated with Wnt molecule expression and AGEs.

We agree with the reviewer that the assessment of biomechanical parameters in our cohort would increase the importance of this study, giving more insights on the effect of downregulation of Wnt signaling on bone strength. Thus, we followed reviewer suggestion, and we performed bone compression tests on trabecular bone core. We found a significant decrease in bone plasticity of T2D compared to controls Young’s Modulus 21.6 (13.46-30.10 MPa) vs. 76.24 (26.81-132.9 MPa); (p=0.0025). We added results of bone compression test in a new paragraph (Page 8, lines 191-194). In order to assess the validity of our results, we performed a post-hoc power calculation using G*Power 3.1.9.7. We demonstrated that effect size was 1.4716626, with a power of 0.9730784, confirming that sample size was sufficient to achieve adequate statistical power. We added methods in the related section and biomechanical data in table 3; we modified the manuscript accordingly (modifications are shown in track changes). Moreover, we also performed correlation analysis between Wnt target genes, AGEs and biomechanical parameters showing significant correlations as reported in the added paragraph in the results section (Page 11, Lines 225-233).

**REVIEWER #2**
This study reports the levels of expression of selected genes implicated in Wnt signaling in trabecular bone from femur heads obtained after surgery from post-menopausal women with (15 women) or without (21 women) type 2 diabetes. They found higher expression levels of SOST and WNT5A, and lower expression levels of LEF-1 and WNT10B in tissues from subjects with T2D, correlating with glycemia and advanced glycation products. No significant differences in bone density were observed. Overall, this is a cross-sectional, observational study measuring a limited set of genes found to vary with glycemia in postmenopausal women undergoing hip surgery.Strengths:The study demonstrates the feasibility of measuring gene expression in post-surgical trabecular bone samples, and finds differences associated with glycemia despite a relatively small number of subjects. It can form the basis for further research on the causes and consequences of changes in elements of the WNT signaling pathway in bone biology and disease.Weaknesses:The small number of targeted genes does not provide a comprehensive view of the transcriptional landscape within which the effects are observed. The gene expression changes are not associated with cellular or physiological properties of the tissue, raising questions about the biological significance of the observations.

We thank the reviewer for the comment. Replying to his/her concerns we have increased the number of Wnt target genes including more interactors of Wnt/β-catenin pathway. We measured GSK3B, AXIN2, BETA-CATENIN and SFRP5 gene expression levels, showing a significant increase in GSK3B, in line with a downregulation of Wnt signaling in T2D. We modified the manuscript accordingly with this new analysis and updated the figure 1 panel (Page 10, lines 210-213). Unfortunately, in this paper we were not able to perform experiments on cellular or physiological properties. However, in order to analyze the biological effect of the analyzed genes on the phenotype, we measured bone strength by performing compression tests on trabecular bone cores (Page 10, lines 201-203 and table 3) and used biomechanical parameters for correlation analysis with targeted genes showing significant correlations of bone strength and Wnt genes. We modified adding a new paragraph in the result section and a new figure panel to the main manuscript (Page 11, lines 225-233 and figure 4).

COMMENTS:(1) The small number of targeted genes does not provide a comprehensive view of the transcriptional landscape within which the effects are observed. Given the author's success in obtaining good-quality RNA from trabecular bone, a more comprehensive exploration would greatly improve the quality of the study.

We agree with the reviewer that increase the transcriptional landscape related to Wnt signaling would be of interest for this work and we really thank for this opportunity. We were able to increase the number of Wnt target genes including more interactors of Wnt/β-catenin pathway, using the same cohort of patients in which we performed the other analysis. We also measured GSK3B, AXIN2, BETA-CATENIN and SFRP5 gene expression levels, showing a significant increase in GSK3B, in line with a downregulation of Wnt signaling in T2D. We modified the manuscript accordingly with this new analysis and updated the figures panel (Page 10, lines 210-213 and Figure 1).

(2) The gene expression changes are not associated with cellular or physiological properties of the tissue, raising questions about the biological significance of the observations. Can the authors perform immunohistochemistry to associate the changes in gene expression with protein expression?

We sincerely acknowledge this comment for focusing the attention on a such important aspect. We have partially replied to this comment in the previous paragraph. Regarding immunohistochemistry analysis, it is not possible to further use the available samples. This is mainly due to the fact that non-decalcified bones were embedded in plastic to allow for separate analysis of newly formed osteoid and mineralized bone. This process leads to poor antigen preservation and unsuitable detection of most targets. Moreover, antibodies for Wnt are also unreliable due to the secreted nature of the protein. Overall, this approach is unlikely to work efficiently. Similarly, RNAscope is not possible due to the resin. Optimization and validation of these analyses will need to be saved for a future study with fresh specimens.

**REVIEWER #3**
The manuscript by Leanza and colleagues explores the regulation of Wnt signaling and its association with advanced glycation end products (AGEs) accumulation in postmenopausal women with type 2 diabetes (T2D). The paper provides valuable insights into the potential mechanisms underlying bone fragility in individuals with T2D. Overall, the manuscript is well-structured, and the methodology is sound. I would suggest some minor revisions to improve clarity.Strengths:The study addresses an important and clinically relevant question concerning the mechanisms underlying bone fragility in postmenopausal women with T2D.The study's methodology appears sound, and the inclusion of postmenopausal women with and without T2D undergoing hip arthroplasty adds to the clinical relevance of the findings. Additionally, measuring gene expression and AGEs in bone samples provides direct insights into the study's objectives.

The manuscript presents data clearly, and the results are well-organized.

Weaknesses:Title. The title could be more specific to better reflect the content of the study. Also, the abstract should concisely summarize the study's main findings, providing some figures.

We thank the reviewer for this suggestion, and we modified the title giving specific information on the main findings of this study. The new title is “Bone canonical Wnt signaling is downregulated in type 2 diabetes and associates with higher Advanced Glycation End-products (AGEs) content and reduced bone strength”. Moreover, we added as suggested a graphical abstract summarizing our study results.

Introduction: the introduction would benefit from the addition of a clearer, more focused statement of the research questions or hypotheses guiding this study.

We thank the reviewer for this opportunity and we reformulated the hypothesis of this study based on our data and new findings as follow:” we hypothesized that T2D and AGEs accumulation downregulate Wnt canonical signaling and negatively affect bone strength”. (page 6, lines 116-117).

Methods: more information is needed on the hystomorphometry analysis. Surgical samples from 8 T2D and 9 non-diabetic subjects were used for histomorphometry analysis. How did these subjects compare with the other subjects in the T2D and control groups? Were they representative? How were they selected?

We thank the reviewer for the opportunity to clarify this important point. The number of subjects included in the different analysis of the paper differ for multiple reasons.In particular, we used only bone specimen with enough trabecular bone material adequate to perform histomorphometry analysis. Therefore, the samples used in the histomorphometry analysis belong to the same subjects enrolled in the study and analyzed for the other experiments of this paper. However, we have previously calculated sample size for bone histomorphometry analysis using the only available data of trabecular bone in T2D postmenopausal women measured by dynamic histomorphometry (Manavalan JS et al, JCEM 2012). We performed a priori sample size calculation using G*Power 3.1.9.7., based on the t-test of two independent groups. Analysis demonstrated that given an effect size of 2.2776769, we needed a total of 12 patients (6/group) to reach a power of 0.978.

COMMENTS:(1) In the Abstract, values and p-values for comparisons, and Spearman's rho and p-values for correlations should be provided. Most adverbs (thus, accordingly, importantly) could be omitted to improve conciseness and clarity.

We kindly thank the reviewers for this precise and careful comment. We changed the Abstract accordingly. According to the abstract style of the journal we initially reported only the main findings. We have now modified providing values and p values as requested. We defer to the wishes of the editor as to the format in which the abstract should be reported.

(2) Result presentation: 25th and 75th percentile should be provided rather than the interquartile range, to better reflect data distribution.

We thank the reviewer for the opportunity to better clarify this part of the results section. We changed the manuscript accordingly.

(3) Estimated glomerular filtration rate should be calculated and provided as a marker of renal function, rather than serum creatinine values.

We thank the reviewer for the comment, and we modify the manuscript accordingly, adding the eGFR values in table 1 and in the result section.

(4) The manuscript should include a statement confirming compliance with the Declaration of Helsinki, considering that human subjects were involved in the study.

We thank the reviewer for the comment. The study was conducted in accordance with the Declaration of Helsinki. Ethics Committee of Campus Bio-Medico University approved the present study. Informed consent was obtained from all subjects involved in the study. (Page 6, lines 134-137).